# Antimicrobial Resistance (AMR) in the Food Chain: Trade, One Health and Codex

**DOI:** 10.3390/tropicalmed4010054

**Published:** 2019-03-26

**Authors:** Anna George

**Affiliations:** 1Centre on Global Health Security, Chatham House, London SW1Y 4LE, UK; anna.george.c@gmail.com or Anna.George@murdoch.edu.au; 2Public Policy and International Affairs, Murdoch University, Murdoch, WA 6150, Australia

**Keywords:** AMR, One Health, food chain, trade, Codex, WHO, World Trade Organization (WTO)

## Abstract

Strategies that take on a One Health approach to addressing antimicrobial resistance (AMR) focused on reducing human use of antimicrobials, but policy-makers now have to grapple with a different set of political, economic, and highly sensitive trade interests less amenable to government direction, to tackle AMR in the food chain. Understanding the importance and influence of the intergovernmental Codex negotiations underway on AMR in the Food Chain is very weak but essential for AMR public policy experts. National and global food producing industries are already under pressure as consumers learn about the use of antimicrobials in food production and more so when the full impact of AMR microorganisms in the food chain and on the human microbiome is better understood. Governments will be expected to respond. Trade-related negotiations on access and use made of antimicrobials is political: the relevance of AMR ‘evidence’ is already contested and not all food producers or users of antimicrobials in the food chain are prepared to, or capable of, moving at the same pace. In trade negotiations governments defend their interpretation of national interest. Given AMR in the global food chain threatens national interest, both AMR One Health and zoonotic disease experts should understand and participate in all trade-related AMR negotiations to protect One Health priorities. To help facilitate this an overview and analysis of Codex negotiations is provided.

## 1. Background: Access to and Use of Antimicrobials

A global political consensus has been reached confirming antimicrobials underpin human health security so access and use of these miracle products has to be wound back across all sectors of the economy [1]. One key agenda slow to emerge is antimicrobial resistance in the food chain with consequences for food safety, food security and significant implications for trade policy. 

The complex integrated strategies needed to reign-in the use of antimicrobials in the food, agricultural and associated industry sectors have the capacity to transform the somewhat benign and logical ‘AMR (antimicrobial resistance) One Health Framework’ into a quagmire of competing interests—as not all producers and users of antimicrobials are prepared to, and some not yet capable of, limiting their use of antimicrobials. But to preserve the AMR One Health global consensus much will depend on how these trade related issues are handled and will require significant leadership and clear recognition of the health security implications of failure.

The UK 2016 O’Neil AMR Report [2] mapped out possible consequences of not safeguarding these precious antimicrobials and analyzed the capacity of this AMR phenomena to economically disrupt and negatively impact on many industry sectors. On the human costs, more accurate research and analysis recording actual numbers of AMR related deaths and the exponential growth of health/productivity costs is emerging that will better reflect the consequences from AMR events [3,4]. This sensitive data is likely to reverberate politically.

Such politically sensitive data and revelations from research on AMR in the food chain will inevitably place ‘food trade’ firmly in the spotlight. Trade policy, at its best, can help induce higher safety standards, better quality food, and safer products. But the spread of AMR microorganisms could represent one of the biggest challenges to trade in safe food and may lead to trade disruption and financial loss. 

Existing trade frameworks and obligations may be capable of addressing this issue but only if sensitized and adapted to prioritize safeguarding health security by providing the flexibility for governments to implement measures to safeguard their food-chain and help preserve antimicrobials, particularly those important for human medicine. 

AMR is an economic and global trade issue, so criticism of government action or regulatory changes perceived as running counter to concepts of ‘free trade’ will need to be managed. But this is also an opportunity for those who have extolled the benefits of trade to step up and deliver on this crucial AMR agenda. 

There are after all several precedents where similar large and economically painful transformations have been deemed necessary linked to ‘access and use’—sometimes for the public or global good and often to facilitate trade in new technologies or facilitate new forms of production and accumulation. For example, intellectual property and copyright provisions extended through WTO Trade-Related Aspects of Intellectual Property (TRIPS) [5]; reducing chemical toxicity in domestic products—EU REACH Legislation (Registration, Evaluation, Authorization and Restrictions of Chemicals) [6], and promoting health objectives—Tobacco Plain Packaging Policy [7]. 

With such complex trade agendas, it was inevitably these transformations created economic disruption by altering access and or use provisions which redistributed costs, benefits, investments and profits. All involved high levels of political contestation as access provisions and/or regulations were redefined at the national level or through multilateral negotiations. Interestingly, lessons and tactics used to support or block these transformations are beginning to resonate in the AMR debates and some have been picked up by media [8].

Implementing effective AMR One Health strategies will require a similar level of political commitment and leadership to transform access and use provisions to preserve the efficacy of antimicrobials, particularly for human medicine. 

## 2. Policy Coherence—Are Trade Policies Understood and Integrated into AMR One Health Strategies?

National AMR One Health Strategies already focus on altering both access provisions and use made of antimicrobials for human use. The other major area of antibiotic use—food production—is yet to be as systematically adapted to achieve national one health objectives. Unlike reducing human use which is negotiated and conducted entirely at the national level usually by government health authorities, but to influence the access and use of antimicrobials in food production (domestic and imported) is more complex. And involves many more interested parties. The 2006 EU ban on the use of antibiotics as growth promotors is an example of the complex legal, trade and political implications that can flow from such decisions.

A fundamental understanding how national trade policies harmonize/comply with international trade rules and obligations is essential. This includes understanding the technical and legal structures that enable and legitimize the use of antimicrobials in the food chain as well as the capacity to exclude them in specific circumstances from imported food. 

This will require AMR public health experts to be active in setting AMR government priorities in these trade-related negotiations to ensure AMR policy coherence: The international standard setting body for food safety, Codex Alimentarius Commission (Codex); the World Trade Organization (WTO); and also Bilateral and Regional Free Trade Agreements.

The rational for engaging in such esoteric areas of trade policy is that any new interpretations, obligations, rules, procedures/guidelines evolving from, for example, Codex negotiations on ‘AMR in the food chain’ have the capacity to impact on the access to and use made of antimicrobials. But of equal importance, such multilateral decisions could also circumscribe the regulatory and legal options available to national governments in implementing their AMR strategies and their domestic export/import policies if national legislation/regulations are not introduced or adapted to reflect One Health priorities. One example to be aware of is introducing the capacity to develop ‘national lists’ of antimicrobials as discussed in Codex TFAMR Report REP 19/AMR. 

An overview of negotiations currently underway in the Codex food standard setting body may help make transparent the complex political and legal obligations linked to international trade. Understanding the food/trade linkage is critical especially if national inter-agency policy cohesion has not fully integrated these trade-related elements. And introducing new regulations on AMR may be problematic if political commitment or bureaucratic capacity to regulate is weak.

For example, the technical and scientific complexity of the AMR/food subject matter and navigating the huge number of Codex standards and guidelines [9] is challenging so these negotiations are usually left to ‘expert’ bodies responsible for Codex, or, decisions driven by broader national trade objectives managed through foreign/trade policy negotiators. 

Bureaucratically integrating the trade agenda into One Health Action Plans may be difficult but is essential. These trade-related linkages should be comprehensively understood for their effect and appropriately responded to in-line with national AMR One Health security priorities. 

## 3. Codex Alimentarius Commission (Codex): Current Negotiations on AMR in the Food Chain and Understanding the Political Context

AMR in the food chain was earlier addressed through the Taskforce on Antimicrobial Resistance (TFAMR) from 2007–2011. In 2016 Governments agreed to re-convene the TFAMR with a broader mandate to address the entire food chain and to report back to the Codex Commission by 2020 [10]. The Terms of Reference are: to revise the *Code of Practice to Minimize and Contain Antimicrobial Resistance* and to develop new *Guidelines on integrated monitoring and surveillance of antimicrobial resistance.*


Gaining consensus agreement through this Codex/TFAMR process may not be easy, particularly as these two documents will also be directly and indirectly endorsing the use of antimicrobials in the food chain. Which antimicrobials can be used in the food chain and in what circumstances represents a highly contested political agenda, particularly antimicrobials used for growth promotion and those deemed essential for human medicine. Codex Guidelines endorsed by Member States may also provide direct or indirect legitimacy for the use of these antimicrobials. 

Unlike WTO negotiations, Codex/TFAMR negotiations enable participation and active input from non-government entities. A reading of the open-source negotiating draft documents with input from governments, industry and consumer representatives provide insights into some of the more contentious areas [11]. While few would argue the need for global collaboration (such as the TFAMR process) to minimize the spread of AMR microorganisms is important but if significant differences arise over containing the use of antimicrobials in the food chain this could in-effect serve to hinder government action to proactively protect consumers. 

The question of consistency with WTO rules is often a good excuse for government inaction. And an added factor to be cognizant of—given the ‘standard setting’ role of Codex which links directly into related WTO obligations—is that Codex standards can, and are, often used to justify positions taken in WTO Trade Dispute cases. Or trade disagreements arising when Sanitary and Phytosanitary (SPS) or Technical Barriers to Trade (TBT) agreements are enacted to restrict or place conditionality on imports.

The use of antimicrobials in the food chain is a politically and scientifically contested agenda. And, despite the UN General Assembly ‘public health security’ framing and political endorsement of the WHO AMR One Health framework and the Global Action Plan (GAP) [12], the Codex/TFAMR parameters open for discussion may not sufficiently prioritize or be consistent with ‘human health’ priorities. Given that a key human health priority is to maintain the efficacy of medically important antimicrobials, so the veracity of action taken to achieve this will be a significant indicator. AMR policy makers should monitor this agenda closely.

For example, to date, neither of the TFAMR draft negotiation texts refer to the WHO Guidelines on Use of Medically Important Antimicrobials in Food-producing Animals biocides appear to now be excluded; and, altering Codex Maximum Residue Limits (MRL) to consider MRLs for medically important antimicrobials do not seem to be open for discussion.

## 4. AMR One Health Policies: Role of the World Trade Organization (WTO)

The WTO along with other international agencies has responded to the United Nations General Assembly Resolution on AMR. WTO Director General, Azevedo, has stated the existing WTO framework provides non-discriminatory measures and flexibility for governments to address AMR One Health policies especially around food safety [13]. Azevedo is stating the obvious—it is government’s responsibility to activate the legal and regulatory framework to protect their citizens. 

AMR in the food chain is a new and complex challenge but implementing such legal and regulatory policies referred to by the WTO DG may not be simply. The international trade environment has expanded considerable since the WTO was established and is more legally complex. National trade policy and the governance framework often have to account for both WTO obligations and broader more intrusive obligations imbedded in new FTAs which may limit the scope for independent national based policy development. 

Also, some important structural and capacity issues may be relevant. For example, governments who have lost some in-house regulatory and governance capacity through adopting neo-liberal market based self-regulation strategies and some regulatory limitations flowing from harmonization and trade facilitation policies. Public health experts are often not sufficiently involved in these trade negotiations. 

Azevedo’s view that the WTO enables implementation of AMR One Health strategies rests on government’s commitment at the national level to manage/protect the domestic and export food chain in line with WTO obligations. For food-related imports the Sanitary and Phytosanitary (SPS) or Technical Barriers to Trade (TBT) agreements can be activated but have a relatively narrow interpretive space unless backed up by national regulations. 

This WTO report to Codex/TFAMR also records individual governments’ input on AMR issues linked to SPS reporting and illustrates some sensitive trade access issues yet to be tested, particularly related to proposed EU regulations [14] (pp. 8–12) The TBT provisions are also likely to be a strong focus as consumers demand of governments more accurate labelling information on antimicrobial use [15].

Implementing longer term SPS measures may rely on specific ‘scientific evidence-based data’ i.e., directly linking food to human transfer of AMR microorganisms. Emergency measures to contain contaminated food imports are generally considered to be short-term temporary measures. In implementing national regulations that are compliant with WTO obligations the key concept is ‘non-discrimination’ (in trade parlance—national treatment provisions). This FAO/WTO ‘toolkit’ is an excellent guide to comprehend these trade rules and obligations for both policy makers and non-WTO experts [16] (pp. 12–17).

## 5. State of Play: Codex TFAMR Negotiations on AMR in the Food Chain

The health concerns linked to AMR in the food chain encompass both the pathogenic and non-pathogenic AMR microorganisms as both can have serious health consequences [17] (pp. 7–10). It is not yet clear how the ‘non-pathogenic’ AMR microorganisms in the food chain will be dealt with in the Codex TFAMR process. 

Those involved in AMR research, media and consumers may be surprised to know that there are major gaps in monitoring/surveillance and proactive testing for AMR microorganisms in the food chain. Only in 2016 was the draft proposal from the specially convened London Meeting forwarded to Codex and integrated into TFAMR’s mandate to develop surveillance guidelines [18] (p. 5). Few if any countries currently systematically test food imports for the presence of AMR microorganisms (whether immediately harmful or not). The rational for lack of action is often circular—based on claims of not enough scientific evidence and/or on the need to first comply with WTO trade obligations [19].

Even in countries with sophisticated governance processes and reliable economic and trade statistics there are considerable gaps in understanding the volume and use made of antimicrobials in animal production and the AMR consequences flowing from this use. 

Antimicrobials used in agriculture and aquaculture production are not well understood and even less is known of the effects of AMR in the environment, wildlife, water, or soil etc. [20] Addressing the largely unknown environmental factors, that link to broader forms of AMR contagion has been particularly slow to receive substantive oversight or policy/regulatory focus [17,20,21]. Only some of these aspects may actually be considered in the Codex TFAMR process.

Always in multilateral negotiations, language, and agreed text describing the terms and definitions of the problem areas and the scope of issues, principles, and definitions that can be legitimately addressed are fundamental. And these definitions will impact on the capacity to agree on meaningful outcomes to address the issues at hand.

The formal intergovernmental negotiations remain non-transparent to the broader public and media but the open-source TFAMR working draft texts to revise the AMR Code of Practice (CRD20) [22] and develop new Surveillance Guidelines (CRD18) [23] are available and convey the complexity and political sensitivity of these negotiations. Most useful in providing a sense of negotiations is the formal reporting prepared for the July 2019 Codex Alimentarius Commission, which synthesizes TFAMR outcomes indicating consensus language and points of difference [24]. Several of these outstanding and contested issues will be worked through intersessionally by the two drafting groups led by US and Netherlands and reported to the next TFAMR negotiations in December 2019.

## 6. Codex/TFAMR Political Sensitivities and Contentious Issues

The current work program of the Codex TFAMR negotiations indicates a considerable amount of work has yet to be undertaken, particularly on the new issues being addressed. It may also be difficult to meet the 2020 deadline. The following three issues are included below as ‘Case Studies’ for those who wish to delve further into the negotiating dynamics. These Case Studies illustrate some of the complex issues yet to be dealt with and deserve the attention and active engagement by governments, consumers, media and public health experts. 

(1)The scope of the ‘food chain’—new issues to be included;(2)Securing antimicrobials of importance for human medicine;(3)Interpretation attached to evidence—scientific evidence-based versus precautionary principle.

## 7. Conclusions

The global transition to safeguard antimicrobials is underway but care will have to be taken to ensure that health security is not derailed by narrow interpretations or vested interests. No doubt, particularly at this stage of the negotiations many of the parameters for discussion have ambiguity built-in and while this might placate some concerns there is always the danger these limitations can become in-built into the decisions eventually evolving from the TFAMR [22] (pp. 3–7).

The many but yet little known consequences of AMR in the food chain will emerge as research efforts intensify and unravel the complex AMR effects on the broader ecosystem, including wildlife, water, and soil. Highly dangerous zoonotic diseases are already impacted by AMR affecting large populations so ongoing threats from zoonotic diseases cannot be neatly compartmentalized or insulated from the effects of AMR microorganisms originating from the food chain [25,26].

Information of actual and possible spread of AMR identified in the Expert Report—including to wildlife, insects, and parasites—are yet to be revealed and some but not all aspects will be examined in the TFAMR discussions. This raises questions of which international organization will take responsibility. Infectious and zoonotic disease experts, with their established links to national security frameworks, should obviously have an interest in how this AMR gap will be addressed. These experts should also be actively inputting into the Codex TFAMR negotiations. 

Other interesting developments are emerging alongside the Codex/TFAMR negotiations. Leadership on AMR policy is evolving from investors, finance industry and some in the food sectors with potential to be a powerful force for change. Their strategies are now in advance of many government policies and also the current approach being taken in Codex TFAMR negotiations. 

And the 73rd UN General Assembly will convene to consider progress made on AMR and the recommendations developed through its Interagency Coordination Group on Antimicrobial Resistance (IACG) [27]. These deliberations should provide a broader overarching model to drive AMR One Health strategies and provide clearer direction to trade-related AMR negotiations such as the Codex TFAMR.

Achieving consensus on a global approach to minimize the spread of AMR is essential but will require significant leadership and incentives to develop the necessary technical capacity to transition away from relying on antimicrobials. But ultimately it is the responsibility of national governments to maintain public confidence in their food chain and to implement governance and regulatory changes needed to address this global health security threat and protect citizens. 

### Case Study 1. Defining the Scope of AMR in the Food Chain More Broadly

The TFAMR tasked the Codex Secretariat [28] (provided by FAO/WHO) to develop ‘scientific advice’ on the scope of AMR in the food chain. The FAO/WHO convened an expert meeting and produced this Summary Report on foodborne antimicrobial resistance—Role of environment, crops and biocides [20]. The primary purpose was to synthesize current scientific literature concerning the transmission of AMR from environmental sources—including from water, soil, wildlife, humans, and equipment. 

This Expert Group Report, distributed in advance of the meeting, initially was not formally registered on the TFAMR Website [11] but the FAO representative summarized some findings under the item: Scientific Advice to Codex [24], (pp. 1–2): Recording widespread reports of AMR bacteria contamination of foods of plant origin, numerous documented outbreaks of AMR foodborne infections traced to foods of plant origin clearly indicate the potential of these products to transmit AMR microorganisms to human contaminated from multiple sources: water, soil, wildlife, humans, and equipment, and that “Steps should be taken to reduce the likelihood of antimicrobial agents and antimicrobial-resistant bacteria entering the environment from agriculture practices and agricultural food production should be protected from environmental sources of contamination.” [24] (p. 1)

Also, reference made to Good Agricultural Practices—to reduce microbial contamination; and, Integrated Pest Management practices to help reduce the need for antimicrobials; on the use of biocides “…there was strong theoretical and laboratory evidence to indicate biocides select for increased resistance to antimicrobials through cross or co-resistance, but empirical evidence is limited” [24] (p. 2). The expert group recommended biocides should be used according to manufacturers’ recommendations. 

Closing off some issues around biocides the Codex/TFAMR Report now records this agreement: “Antimicrobials used as biocides, including disinfectants, are excluded from the scope of these guidelines” [24] (p. 10).

Some issues raised in the Expert Group Report [20] will be addressed at the next TFAMR meeting in December 2019 and other elements now integrated for further consideration. For example, the draft Code of Conduct definition of ‘the food chain’ was endorsed by the TFAMR as: *“Production to consumption continuum including, primary production (food producing animals, plants/crops), harvest/slaughter, packing, processing, storage, transport, and retail distribution to the point of consumption”* [22] (p. 4). 

Many very sensitive issues, including defining principles are yet to be settled—use of growth promotors and the introduction of government’s developing ‘national lists’ could be usefully developed (but concerns expressed at the potential of such ‘lists’ to impact trade) [24] (p. 6).

Also worth noting is the WHO/FAO/OIE Report to the TFAMR contains a long list of forthcoming expert meetings to research/analyze outstanding AMR issues including many raised by the Expert Group [20]. But this will be a lengthy process before relevant data and advice will be available [29]. 

Given the threat to public health of AMR already affecting the food chain and the broader environment, any delay in taking counter-measures to actively protect citizens from such exposure is highly problematic. Especially if reasons for inaction are predicated on the basis that ‘evidence’ is not available when screening and testing has not been actively pursued by governments or the food industries responsible, or, data is not made transparently available for research.

### Case Study 2: Securing Antimicrobials of Importance for Human Medicine—The WHO CIA List

The WHO has already defined the *List of Critically Important Antimicrobials for Human Medicine* (WHO CIA List) [30] which ranks antimicrobials used in human medicine based on two criteria—importance to human health and the likelihood of resistance transmission through the food chain. The WHO also developed and released what could be described as guidance for implementing this CIA List—*The WHO Guidelines on Use of Medically Important Antimicrobials in Food-producing Animals* (WHO Guidelines) [31].

Given the logical progression of these two WHO documents, which essentially provides important implementation guidance to help preserve the antimicrobials most important for human health, but this appears to be a step too far for some countries not yet ready or prepared to take these steps. This resistance was reflected in the Codex/TFAMR documents which excludes any endorsement of these WHO Guidelines. This is an important issue that will not simply disappear, so some background may be useful.

In 2017 after a two year process the WHO Guidelines were released and immediately drew criticism from the U.S. including in this media release from USDA Acting Chief Scientist questioned the legitimacy of the ‘evidence’ underpinning them as well as the role of the WHO in developing guidelines over subject matter perceived as being the preserve of the Codex and the OIE [32]. This information document from the WHO clarifies the background to the development of the WHO Guidelines and reiterating its role is to protect public health and the antimicrobials important for human medicine. Antibiotics used only in Animals were not included in the WHO Guidelines [33]. 

Some business-focused media coverage provided this commentary on the politics behind this unusual public criticism of the WHO’s mandate to develop such guidance [34]. A later contrary response from some key US Lawmakers on the Codex/TFAMR negotiations regarding the use of ‘growth promoters’ demonstrated the level of internal contestation that can arise [35].

This difference in opinion over the WHO Guidelines was carried through to Codex TFAMR negotiations with the WHO representative being asked to clarify the ‘status’ of the WHO CIA List and its Guidelines. A summary of WHO’s response is below but the full explanation should be understood as it clearly defines the WHO’s mandate to develop these two reports, the governance and operational procedures underpinning them, and the political flexibility accorded to governments [24]. 

The WHO’s statement appears to clarify that both WHO documents have the same status and includes the following points: Both reports are science based, the primary focus is to protect public health and they are not open to negotiations. Their adoption by the World Health Assembly is not required under WHO rules and implementation by Member States is voluntary [24] (p. 2).

With the WHO Guidelines now a source of political contention and questioning the legitimacy of decision making will prove disruptive. But in this important health security agenda questioning the legitimacy of data also has the potential to create a significant fracturing of the global consensus on AMR One Health Policy. Consumer and Health non-government organization’s input to TFAMR indicated their full support for the WHO Guidelines. 

This dispute over politically endorsing the WHO Guidelines will not be resolved easily or quickly as it signals implementing these WHO Guidelines to preserve the WHO CIA List may be politically problematic or too difficult for some countries. 

However, in stark contrast, the recommendations outlined in the OIE List of Antimicrobial Agents of Veterinary Importance [36] and the WHO CIA List [30] are being simultaneously supported. But there are obvious significant compatibility problems that run counter to the objectives of the WHO CIA List. Endorsement of the OIE List sanctions the use of many of the medically important antimicrobials listed in the WHO’s CIA List. 

The two documents may be individually internally consistent according to the guidance for developing them, but not compatible for delivering the objective of preserving the effectiveness of medically important antimicrobials for human use—the WHO CIA List. 

This, of course, is not the only difference in approach, and it would be naïve to expect that such political differences would not arise when significant economic interests are at stake. But questioning the legitimacy of the WHO Guidelines, particularly by such a powerful player as the U.S. could put a break on measures to reduce using medically important antimicrobials that are currently extensively used in food production. Other interested parties may welcome this dispute to delay transitioning away from antimicrobial use. Worth noting, the TFAMR has not yet substantially focused on antimicrobials important for humans also used in crop production or the broader environment [20,37]. These issues will also be highly relevant for zoonotic and infectious disease experts.

Interestingly, asset managers of large investments in the global food industry are moving well ahead of the deliberations in Codex (and many governments). Their agenda links into the WHO CIA list and supports many of the implementation elements contained in the WHO Guidelines [38]. These corporate bodies are aware and expecting AMR trade regulations to be enacted [39] to preserve antimicrobials important for human health. The EU being the most advanced and its One Health Strategy includes commitment to act to protect citizens, food producers and that the efforts made by EU farmers “…are not compromised by the non-prudent use of antimicrobials in EU trading partners” [40]. The U.S. FDA Strategy for the Safety of Imported Food also indicates a strategic focus on consumer safety [41].

### Case Study 3: The Political Agenda: Scientific-Evidence Based Data versus Precautionary Principle

For a complex subject such as ‘AMR in the food chain’ the interpretation of what constitutes ‘evidence’ and the legitimacy this conveys matters—particularly in Codex [42], OIE [43], and the WTO [44] trade-related deliberations. To state the obvious, scientific evidence-based data matters but there are numerous examples of scientific evidence-based claims being overturned as so narrow to be almost meaningless or totally unjustifiable, including many attached to controversial health and food issues i.e. tobacco use, and obesity issues. 

AMR also shines a light on the need to implement and develop basic hygiene and public health infrastructure. Developing countries’ technical capacity/resources to minimize the dangers of AMR in the food chain are yet to be sufficiently addressed [45]. From an economic and development perspective, those countries relying on export earnings from food production are particularly vulnerable. But for those with well-developed public health systems there remains considerable resistance to transparently collect or test the basic data needed to analyze consequences of antibiotic use in their food producing animals and agriculture. 

A reading of the many submissions made into the TFAMR negotiations by government, industry and consumer representatives should leave the reader in no doubt of the underlying sensitivities and interpretations of ‘valid’ scientific data and risk. Some of these positions may however be overtaken by other events. For example, the WTO Secretariat’s Report to the TFAMR demonstrates that multilateral dialogue on trade and AMR in the food chain is being opened up to further scrutiny outside of Codex. WTO Members engaged in a substantive dialogue on AMR issues in the SPS Committee for the first time, primarily focused on EU legislative intentions to address AMR in the food chain [14]. 

This EU regulatory information provided is important and covers a range of issues and given the response from several countries will be politically sensitive and played out in both Codex and WTO forums. Topics worth noting being developed by the EU include legislative measures: Addressing public health risk of AMR; reserving certain antimicrobials for treatment of infections in humans only; misuse of antimicrobials in medicated feed for prophylaxis and limiting treatment duration. The report records interesting responses and questions to the EU representative from several governments. The report also includes a list of ‘regular and emergency’ SPS and TBT Notifications submitted by Member States. 

The debate opened up in the WTO SPS Committee may not yet have fully registered at the December Codex/TFAMR meeting but is significant. These new inputs now formally expressed to the SPS Committee illustrate further the importance of fully integrating WTO and Codex policy into national AMR One Health strategic planning. 

For an observer of the Codex/TFAMR negotiations it is interesting to note that national-based AMR One Health implementation policies are actively reducing human access to antimicrobials. And at the global level, governments have politically endorsed the position that antimicrobials need to be protected and treated as a global public good. Contrasting this, reaching agreement on action to stop or reduce the non-therapeutic use of antimicrobials for food-producing animals and also to preserve medically important antimicrobials for humans, seem to require a much higher standard of scientific evidence-based data. As consumers’ understanding of the AMR One Health agenda develops, they may not support such reticence to act on this important health security issue.

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
