# Peer review of "Antimicrobial Resistance (AMR) in the Food Chain: Trade, One Health and Codex"

_tropicalmed, 2019, doi:10.3390/tropicalmed4010054_

Reviewer 1 Report

This reviewer has respect for the task attempted here. This is a necessary and highly relevant debate, that needs to go outside the narrow Codex discussion chambers.

I have a significant number of comments and questions, that I would hope the author is able to attend to 

In a very general sense I feel the author tends to stay within the narrow Codex negotiation sphere without really referring to the significant amount of national legislation already instituted. All Codex negotiations should also be seen within the context of existing national legislation.

For example the legislation banning the use of AM growth promoters in animals in EU is only mentioned in passing, while the banning of Vet's profit from selling AM in Scandinavia is not mentioned at all. It would seem important for the reader to at least understand that such legislation already exists (and has been shown to be efficient!). Also note that USA and China are in the process of implementing legislation to (partially) deal with the use of AM growth promoters.

Line 53

Comments – Review

Anna George Commentary: Antimicrobial Resistance (AMR) in the food chain: Trade, One Health and Codex

1.       Line 11: “…consequences for food security and significant implications for trade policy.” It is suggested that food safety should be included before food security Line 22

2.       Line 22:  “…. the exponential growth of…”  the authors of the paper referenced do not use the words exponential growth at all in the paper!

3.       Line 23:   “….better reflect the consequences from AMR events.”   Ref. 4 referrred to here is a News report and should not be treated as a peer review documentation of statement!

4.       Line 57: “  National trade frameworks and the international bodies that legitimise or promote the use of antimicrobials in the food chain must also be fully integrated into One Health Action Plans.” It is not clear which international bodies legitimise or promote the use of antimicrobials in the food chain – does this statement suggest that antimicrobials should not be used in food production animals at all? Please clarify!

5.       Line 66: “  And, also capable of defining whether food products linked to the use of antimicrobials can be traded in line with domestic import regulations.” Not at all clear what this means -please clarify!

6.       Line 85: “Analysing the effects of AMR microorganisms in food has been a slow-burning Codex issue, primarily addressed by a Taskforce on Antimicrobial Resistance”. Codex bodies do typically not investigate effect of foodborne hazards; in line with the principle of separating risk assessment and management, effect analyses is performed by FAO/WHO expert bodies. Several such FAO/WHO and FAO/WHO/OIE expert meetings have already previously been convened, and Case Study 1 in this paper refers specifically to the latest FAO/WHO expert meeting. .

7.       Line 107: “Or trade disagreements arising when WTO food safety provisions are enacted by national governments – i.e. the WTO Sanitary and Phytosanitary (SPS) or Technical Barriers to Trade (TBT) provisions.” Not clear what this sentence refers to, please clarify? And not clear what is meant by WTO food safety provisions – WTO SPS and TBT agreements provide frameworks for national food safety provision, and suggests that countries utilize Codex standards and provisions in this work. Which WTO food safety provisions does this sentence refer to?

8.       Line 112: “…the parameters open for discussion may not sufficiently prioritise or be consistent with ‘human health’ priorities.” Needs clarification – which parameters ? is the author suggesting that certain parameters linked to health are specifically excluded from the discussion, if so: please state this clearly and mention the parameters.

9.       Line 121: “  neo-liberal market based self-regulation strategies or policy primarily informed by trade lawyers and complex FTAs obligations rather than public health experts.” What does market-based self-regulation refer to here? Is the author referring to the (relatively) new ‘own-control’ systems of food industries, and if so, how would these affect national AMR regulation? This reviewer is not aware of any national AMR regulatory system that is based on self-regulation? There are, however, in many countries efforts to involve farmers and veterinarians in voluntary systems to reduce AM use (e.g. Netherlands). Is this what the author is referring to, then it should be clearly stated and discussed – for instance in a comparison between the effect of such voluntary action and direct regulatory action (as occurs in most Scandinavian countries).

10.   Line 126: Please clarify which ‘WTO provisions” the author refers to? The author seems to continue to suggest that WTO has specific food safety provisions, this really needs clarification throughout the paper. The SPS Agreement sets out the basic framework for rules for food safety and animal and plant health requirements. It allows countries to set their own standards. However, it also specifies that regulations must be based on scientific findings and should be applied only to the extent that they are necessary to protect human, animal or plant life or health; they should not unjustifiably discriminate between countries where similar conditions exist. There are no specific SPS standards, which is also why the SPS agreement specifically refers to Codex and OIE as the relevant international suggested standards in the area.

11.   Line 131: “ more accurate labelling information a microbial use.” I believe the author means ‘information on antimicrobial use.’

12.   Line 132: “Long-term SPS measures rely on specific ‘scientific evidence-based data’ i.e. directly linking food to human transfer of AMR microorganisms. And emergency measures to contain contaminated food should only be temporary.” Which long-term SPS measures is the author referring to here? And what is the link to emergency measures. Typically long-term measures would be regulatory measures instituted by the country in question. There are a significant number of AMR national regulatory measures based on scientific evidence, where the effect of such measures has also been documented (see Wielinga et al., 2014 and Dar et al, 2016), and to my knowledge none of these have yet been contested under the WTO?

13.   Line 139: “food poisoning event managed through WTO SPS provisions” Food poisoning events are not managed through SPS provisions – countries have food control management systems, typically based in national food laws, and these systems also govern the management of foodborne disease events.

14.   Line 147: “Few if any countries currently systematically test food imports for the presence of AMR microorganisms” Several countries actually have had functioning AMR surveillance systems for more than 20 years. The main reason not to use specific import testing is that this is a very in-efficient way to improve food safety in a country. You need data from all foods, including imported and own-grown, which is why efficient surveillance systems look into both.

15.   Line 151: an “in” missing

16.   Line 187: known in stead of know

17.   Line 259: WTO should be replaced with WHO

18.   Line 337: “..to privilege evidence based-data and undervalue precautionary based decision making.” The author seems to suggest that precautionary decision making does not relate to evidence – it should be noted that the application of precaution relates to the evaluation of uncertainty related to specific (scientific) data, and therefore the difference in approach is really what you should do when you have significant uncertainty: one school favours doing nothing and the other (precautionary) school favours doing something, while also attempting to get more evidence (and thereby lower the uncertainty).

19.   Line 344: “ … that national-based policies already supported and actively reducing humans accessing antibiotics for medical reasons can be easily justified on the basis of the precautionary principle” The fact that reducing use of AM in humans reduces AMR is a fact that does not need qualification with the precautionary principle. (see Dar et al, 2016)

References:

Dar et al (2016): Exploring the evidence base for national and regional policy interventions to combat resistance. Lancet Vol 387, Jan 16, 2016

Wielinga et al. (2014) Evidence-based policy for controlling antimicrobial resistance in the

food chain in Denmar. Food Control 40 (2014) 185-192

Author Response

Editor: To assist in checking the input I have highlighted the changes made in italics.  The new page numbers have been added in square brackets at the end of the proposed text. And highlighted information for the Reviewer as Comment in response to extra explanatory information.  

Reviewer: Individual comments are addressed below.  Regarding the scale of the task, my intention has been to attempt to raise the understanding and significance of Codex’s role and specifically that of the TFAMR intergovernmental negotiations in developing outcomes impacting on national and global AMR policy. The primary aim being to encourage those with responsibility for AMR One Health policy to understand the nexus between AMR and Trade; the role and impact of the multilateral institutions; and, touching only briefly on, the importance of also awareness of any constraints evolving from FTAs.  And, to argue the case strongly that all AMR trade-related negotiations should be understood for their effect and capacity to influence the health security implications of AMR. And that national trade policy be cohesively integrated into One Health Policy and Implementation strategies.  I acknowledge the value and progress made, particularly by EU policy institutions and some committed Members to develop AMR health security policy and food safety in general but these aspects could not be fitted into this Commentary which is already a significant size. Appreciate the detailed attention given to the review.

1.    Line 11. Accept – ‘food safety’ added

2.    Line 22:  The sentence will be altered to place reference 3 at the end of the sentence.  Reviewer is correct the authors did not use the word ‘exponential growth’ this was my description and was not signalled as a direct quote.  [23-24]

Comment:  I included the Burnham reference as it indicates a significant gap in analysis and data collection regarding deaths. This earlier Bureau of Investigative Journalism 2016 article revealed the failure of health officials to accurately reflect MDRO related deaths.  Last year Dame Sally Davies in her evidence to the UK Parliamentary Committee on AMR raised this gap in policy as being significant and must be addressed. The reporting on AMR deaths has been problematic (23,000 for the US 25,000 for the EU) for some years given the growth of MDRO infections. However, these figures (often sourced from the O’Neil Report) continue to be quoted in many peer reviewed journals and media. In part this reflects a failure of health systems to more accurately address the ‘costs’ of AMR, and also of One Health policy governance to more actively collect data.  Perhaps the broader question to be addressed, not attempted in this article, is why is this the case. 

3.    Line 23:  Editor’s Attention:Footnote 4. I defer to the Editor to interpret Journal policy on Commentary Articles. My preference is to retain but the salient point does not rely on this reference only that it would very much help the reader to understand the context and background. This CIDRAP reference records: A relevant case study; interview with and background to Burnham’s article; comments from CDC officials on the 2013 CDC calculations of 23,000 US deaths; and, information on CDC plans to release new estimates of illnesses and deaths caused by drug-resistant pathogens later this year.   The Centre of Infectious Disease and Research Policy (CIDRAP) is a respected source of up to date information on AMR and infectious diseases and useful for the non-scientist or AMR expert.

4.    Line 57.  There is no suggestion that antimicrobials should not be used in food producing animals. This revised input replaces second sentence (line 57-8) and explains in more detail.

“The other major area of antibiotic use – food production - is yet to be as systematically adapted to achieve national one health objectives.  Unlike reducing human use which is negotiated and conducted entirely at the national level usually by government health authorities, but to influence the access and use of antimicrobials in food production (domestic and imported) is more complex. And involves many more interested parties.  

Understanding how national trade policies harmonise/comply with international trade rules and obligations is essential. This includes understanding the technical and legal structures that enable and legitimise the use of antimicrobials in the food chain as well as the capacity to exclude them in specific circumstances from imported food. One example of the legal and political implications is the 2006 EU ban on the use of antibiotics as growth promoters.”

5.    Line 66-68: Delete sentence “And also capable…in line with domestic import regulations”.  

Line 70: add after export/import policies “if national legislation/regulations are not introduced or adapted to reflect national One Health priorities. One example to be aware of is introducing the capacity to develop ‘national lists’ of antimicrobials as discussed in Codex TFAMR Report REP19/AMR [Editor Please add reference to f/note 24]  [76-78]

6.    Line 85:  replace sentence with. ‘AMR in the food chain was earlier addressed through the Taskforce on Antimicrobial Resistance (TFAMR) from 2007-11.”   [93-94]

7.    Line 107.  The specific phrase ‘food safety provisions’ deleted - is not found in the WTO Agreement.  Reference altered to: 

Or trade disagreements that can arise when Sanitary and Phytosanitary (SPS) or Technical Barriers to Trade (TBT) agreements are enacted to restrict or place conditionality on imports.’  [126-127]

Comment/explanation: Terms such as WTO food safety provisions / WTO TRIPS access to medicines provisions/Para 6 provisions etc are often used as shorthand in negotiations referring to measures or agreements such as SPS/TBT operating procedures which are fundamental in providing the political balance between ‘free’ trade and food safety in the WTO Agreement.

8.    Line 112. Reviewer questions ‘Parameters’: 

Given that a key human health priority is to maintain the efficacy of medically important antimicrobials, assessing the veracity of action taken to achieve this will be a significant indicator. AMR policy makers should monitor this agenda closely.

For example, to date neither of the TFAMR draft negotiating texts refer to the WHO Guidelines on Use of Medically Important Antimicrobials in Food-producing Animals; [Insert cross ref 31]biocides appear to now be excluded [ref 24 p.2 and p.10];and, altering Codex Maximum Residue Limits (MRL) to align for example with possible actions linked to preserving medically import antimicrobials, do not seem to be open for discussion.

   And, despite the UN General Assembly ‘public health security’ framing and political endorsement of the WHO AMR One Health framework and the Global Action Plan (GAP), the Codex/TFAMR parameters open for discussion may not sufficiently prioritise or be consistent with ‘human health’ priorities. Codex/TFAMR intergovernmental negotiations are to provide guidelines to governments to address AMR in the food chain.  Given a key human health priority is to maintain the efficacy of medically important antimicrobials so the veracity of action taken could be a significant indicator. AMR policy makers should monitor closely. [132-139]

Comment:I recognise that opening up this area would be very difficult but is nevertheless relevant, particularly if countries impose bans on the use of particularly antibiotics. Please note, the reference to MRLs has been raised by Brazil in relation to proposed EU legislation see page 10 para 1.20 of CX/AMR 18/6/4

9.    Line 121.      AMR in the food chain is a new and complex challenge but implementing such legal and regulatory policies referred to by the WTO DG may not be simply.  The international trade environment has expanded considerable since the WTO was established and is more legally complex. National trade policy and relevant governance structures often have to account for both WTO obligations and broader more intrusive obligations imbedded in new FTAs which may limit the scope for independent national based policy development.

Also, some important structural and capacity issues may be relevant. For example, for governments who have lost some in-house regulatory and governance capacity through adopting neo-liberal market based self-regulation strategies and some regulatory limitations flowing from harmonisation and trade facilitation policies. Public health experts are often not sufficiently involved in these trade negotiations. [146-156]

Comment: Given the Reviewer appears to have several concerns over my reference to: “AMR in the food chain is a new and complex challenge particularly for governments who have lost some regulatory capacity by opting for neo-liberal market based self-regulation strategies or policy primarily informed by trade lawyers and complex FTA obligations rather than public health experts.”  Hopefully the following information to both clarify my intention and provide some justification to the comment made will suffice.

Firstly, I do not consider that my original text claimed that ‘any national AMR regulatory system will be based on self-regulation.’  I was pointing to the complexity and practicalities of minimising AMR in the international food chain given factors such as the contemporary, post Doha trade environment and the significant political shift towards self-regulation, fee for service, outsourcing policy and scientific assessments etc. which is well understood as very much, part of and complementary to, a neo-liberal policy framework. It would be remiss to ignore the balance of interests involved in such arrangements which may impact on government’s commitment or capacity to regulate. I did not refer to ISDS provisions, but it is an highly political element that has a particular impact on action to altering any trade related or investment policy. There are plenty of examples of chilling effect of signing on to ISDS Agreements.

This Commentary piece is not the place to broaden out the debate on the prevailing political/economic environment, or to include some of the positive voluntary and private/government collaboration identified by the Reviewer. The purpose is to provide a measure of understanding of the current Codex Intergovernmental TFAMR negotiations on AMR in the food chain; illustrate the nexus between domestic and international trade policy obligations; and, also illustrate the trade-policy capacity needed to understand constraints that can be encountered in implementing national One Health strategies if a comprehensive understanding of ‘trade’ is not integrated into national One Health Strategies. 

The reference to neo-liberal market … is relevant and provides context to the trade and policy operational/enforcement environment which for many countries, including those who have, since the Doha Round, negotiated new Bilateral and regional FTAs. The generally focus on harmonisation to a particular regulatory and legal model and a trade facilitation process that recognises WTO SPS and TBT Agreements but has added new enhanced obligations FTA Parties now adhere to.

These FTA ‘behind the border’ processes move away from the WTO approach to implementing WTO obligations which is both defined and implemented at the national level. Whereas, and in contrast, some FTAs now include the ‘other party’coming directly into and participating in national policy negotiations  and/or regular participation in specific FTA Committees to address SPS/TBT with ‘interested parties’ which can include expert input from the corporate sector.  These legal commitments can put a break on governments acting wholly in the national interest and cannot be compared to for example EU single market arrangements.

10.  Line 126:  Issue clarified in para 7 above.  Take out ‘food safety provisions’ and replace with ‘Agreements’.  [168]

11.  Line 131.’on antimicrobial’use – agree thank you.  [173]

12.  Line 132. Implementing longer term SPS measures may rely on specific …. And emergency measures to contain contaminated food imports are generally considered to be only temporary.  [174-176]

Comment:SPS Annex B uses the term ‘urgent problems…’ but also in other WTO reports  ‘emergency measures’ is used as a descriptor which for the non-expert usefully describes the measure.  See input at number 18 below - Codex CX/AMR 18/6/4 ‘emergency measures’ is used in the WTO official report to Codex.   

13.  Line 139.  Remove first sentence and Add to the beginning of the new first sentence  “The health concerns linked to”  [182]

14.  Line 147: No alteration.  [186-188]

Comment: The review provide interesting background info on some countries, perhaps mostly in the EU. And, of course the EU will be moving to institute changes to its import legislation when its AMR policies and Action Plan have EU wide compliance, particularly in some member states yet to make significant progress in managing AMR (f/n40). 

The comment on ‘a functioning’ AMR surveillance systems for more than 20 years’ and that specific import testing is a ‘very in-efficient way to improve food safety’ is interesting.  This is a position that I often encounter in policy discussions. It is a position that requires much more attention than could be addressed in this Commentary article.

However, inefficient or not such testing may be, until there is a significant commitment to monitor and test imports there will not be sufficient ‘evidence’ collected to assess the spread of AMR through imported food.  Picking up AMR microorganisms from health events are only a part of the agenda, AMR in the Gut microbiome must also be of concern and this area is not sufficiently being addressed. Testing food imports for AMR microorganisms would involve having similar treatment applied to domestic products in line with WTO ‘national treatment’ provisions. From a consumer perspective this would seem to be a positive and confidence building outcome. 

Proactively testing the food chain will likely have consequences for some food producing countries but trade compliance measures are already in place, some managed through special export production zones to comply with for example, EU food legislation.  In the Codex/TFAMR draft documents there are several references citing trade concerns which will have to be carefully addressed.

And lastly, I must concur that  the EU monitoring institutions  ECDC,EFSA, EMAand the EARS-Net makes a significant contribution to AMR surveillance within the EU providing an important leadership role collecting and the dissemination of important data. 

15.  Line 151. Even  ‘in’countries   [193]

16.  Line 187 Take out know and insert ‘known’     [243]

      17.  Line 259  Take out WTO and insert ‘WHO’     [319]

18.  Line 337 and 344.  The Reviewer’s comments are appreciated, and the following alterations made with relevant background information. 

Line 336-9 – delete and replace with: 

A reading of the many submissions made into the TFAMR negotiations by government, industry and consumer representatives should leave the reader in no doubt of the underlying sensitivities and interpretations of ‘valid’ scientific data and risk.  Some of these positions may however be overtaken by other events. For example, the WTO Secretariat’s Report to the TFAMR demonstrates that multilateral dialogue on trade and AMR in the food chain is being opened up to further scrutiny outside of Codex. WTO Members engaged in a substantive dialogue on AMR issues in the SPS Committee for the first time in July 2018, primarily focused on EU legislative intentions to address AMR in the food chain.         [please insert cross-ref 14]

This EU regulatory information provided is important and covers a range of issues and given the response from several countries will be politically sensitive and played out in both Codex and WTO forums.  Worth noting the legislative focus includes: Addressing public health risk of AMR; reserving certain antimicrobials for treatment of infections in humans only (yet to be defined); misuse of antimicrobials in medicated feed for prophylaxis and limiting treatment duration.  The Report records interesting responses and questions to the EU representative from several governments. The Report also includes a list of ‘regular and emergency’ SPS and TBT Notifications submitted by Member States.  

 The debate on AMR and Trade now opened up in the WTO SPS Committee may not yet have fully registered at the December Codex/TFAMR meeting but is significant. These new inputs now formally expressed to the SPS Committee illustrate further the importance of fully integrating WTO and Codex policy into national AMR One Health strategic planning.    [408-426]

19.  Line 344: Take out: ‘can be easily justified on the basis of the precautionary principle’ 

Line 347 add Contrasting this, “reaching’   [430]

Add at the end of the sentence “Consumers are unlikely to come to share that view.” [432-433]

Reviewer 2 Report

No comment. The paper is well written, well researched and of high interest to those in the AMR field

Author Response

Thanks

Round  2

Reviewer 1 Report

Have no more concerns with the paper

Still don't agree that border control is the best solution - but I don't need to agree with everything the author says!